**Data Availability Statement:** All relevant data are within the manuscript and its Supporting Information files.

# Effect of enforcement of the national referral guidelines on patterns of orthopedic admissions to Kenyatta National Hospital, Kenya: Pre-post intervention study

**Maxwell Philip Omondi**[1]*, **Joseph Mwangi Chege**[1], **Herbert Ong'ang'o**[2], **Fred Chuma Sitati**[1]

**1** Orthopedics Unit, Department of Surgery, University of Nairobi, Nairobi, Kenya, **2** Kenyatta National Hospital, Nairobi, Kenya

* maxwellomondi@gmail.com

## Abstract

### Background

Inappropriate utilization of higher-level health facilities and ineffective management of referral processes in resource-limited settings are becoming increasingly a concern in health care management in developing countries. This is characterized by self-referral and frequent bypassing of the nearest health facilities coupled with low formal referral mechanisms. This scenario lends itself to a situation where uncomplicated medical conditions are unnecessarily managed in a high-cost health facility. On July 1, 2021, Kenyatta National Hospital (KNH) enforced the Kenya Health Sector Referral Implementation Guidelines, 2014, which required patients to receive approval from the KNH referral office and a formal referral letter to be admitted at KNH to reduce the number of walk-ins and allow KNH to function as a referral facility as envisioned by the Kenya 2010 Constitution and KNH legal statue of 1987.

### Objective

To determine the effect of enforcing the national referral guidelines on patterns of orthopaedic admissions to the KNH. This was a pre-post intervention study. Data abstraction was done for 459 and 446 charts before and after the enforcement of the national referral guidelines, respectively.

### Results

Enforcement of the national referral guidelines reduced the proportion of walk-in admissions from 54.9% to 45.1%, while the proportion of facility referrals increased from 46.6% to 53.4% (p = 0.013). The percentage of non-trauma orthopaedic admissions doubled from 12.0% to 22.4% (p<0.001). There was also an increase in admissions through the Outpatient Clinic and Corporate Outpatient Clinic. The proportion of emergency admissions declined, while that of elective admissions increased. The increase in elective cases was mainly driven by the increase in female admissions with active insurance cover, tertiary

**Funding:** This study was partially funded by Kenyatta National Hospital (https://knh.or.ke/). MPO received the funding award. The research grant award was RFA 2021/22. The funders had no role in study design, data collection and analysis, decision to publish, or preparation of the manuscript.

**Competing interests:** The authors have declared that no competing interests exist.

education, non-trauma-related conditions and older age groups. However, the use of official formal written referral letters did not change despite the enforcement of the national referral guidelines.

## Conclusion

The enforcement of the national referral guidelines reduced the proportion of walk-ins' admissions to KNH. While the enforcement of the national referral guidelines had no effect on the use of official formal written referral letters, it did limit access and utilization of inpatient orthopedic services for young male patients with no active insurance cover and in need of emergency orthopedic care.

## Background

Kenya is administratively organized into forty-seven (47) counties with the Nairobi Capital city being in Nairobi County [1]. Health care service delivery has been devolved to the county level with the national level responsible for health policy formulation and strategic planning. Kenyatta National Hospital (KNH) is managed by the national government while County and Sub-county level hospitals are managed by the County government administratively [1, 2]. The Kenya health service delivery system is organized across six levels of care, beginning at the community level (level 1) and continuing through primary care services, which include dispensaries (level 2) and health centers (level 3), and county referral health services (level 4 and 5) all the way to the national referral health services (level 6) (Fig 1) [2]. The KNH is a level 6 facility and offers very specialized skills, expertise and services [2].

KNH was established in 1901 to provide training and medical research in Kenya. It became a State Corporation in 1987 and sits at the apex of the health referral system in Kenya [3]. According to the KNH Board order of 1987 contained in Legal Notice No. 109, the functions of KNH were spelt out as a) to receive patients on referral for specialized health care; b) to provide facilities for medical education for the University of Nairobi and other health allied courses; and c) to contribute to national health planning [3]. This understanding has been reinforced by the Kenya Health Sector Referral Implementation Guidelines, 2014, and the Constitution of Kenya 2010 which also tasks the KNH with the responsibility for health policy formulation [1, 2].

Tertiary hospitals in resource-limited countries treat referred patients but in most cases are the first level of care for the vast majority of patients [4]. One of the challenges in health care delivery in resource-limited settings is the inappropriate utilization of tertiary health facilities which results in patient congestion with simple conditions that can be effectively managed at lower health facilities. The majority of these patients are self-referred, bypassing lower-level health facilities during the process [5–8]. This jeopardizes the appropriate delivery of primary, secondary, and tertiary health care.

Tertiary health care is overwhelmed by the large demand for primary health care that should be managed at lower level health facilities and in the process cripples the primary health care system and this effectively ensures that primary health care facilities remain underused and inefficient [9].

Orthopaedic wards in the KNH have consistently recorded the highest bed occupancy percentage over the last couple of years. In 2018, 2019, and 2020, bed occupancies were 142.2%, 138.2%, and 116.5%, respectively, compared with overall KNH bed occupancies of 106.2%, 113.4% and 91.5%, respectively [10]. This coupled with inadequate staffing levels, results in a low

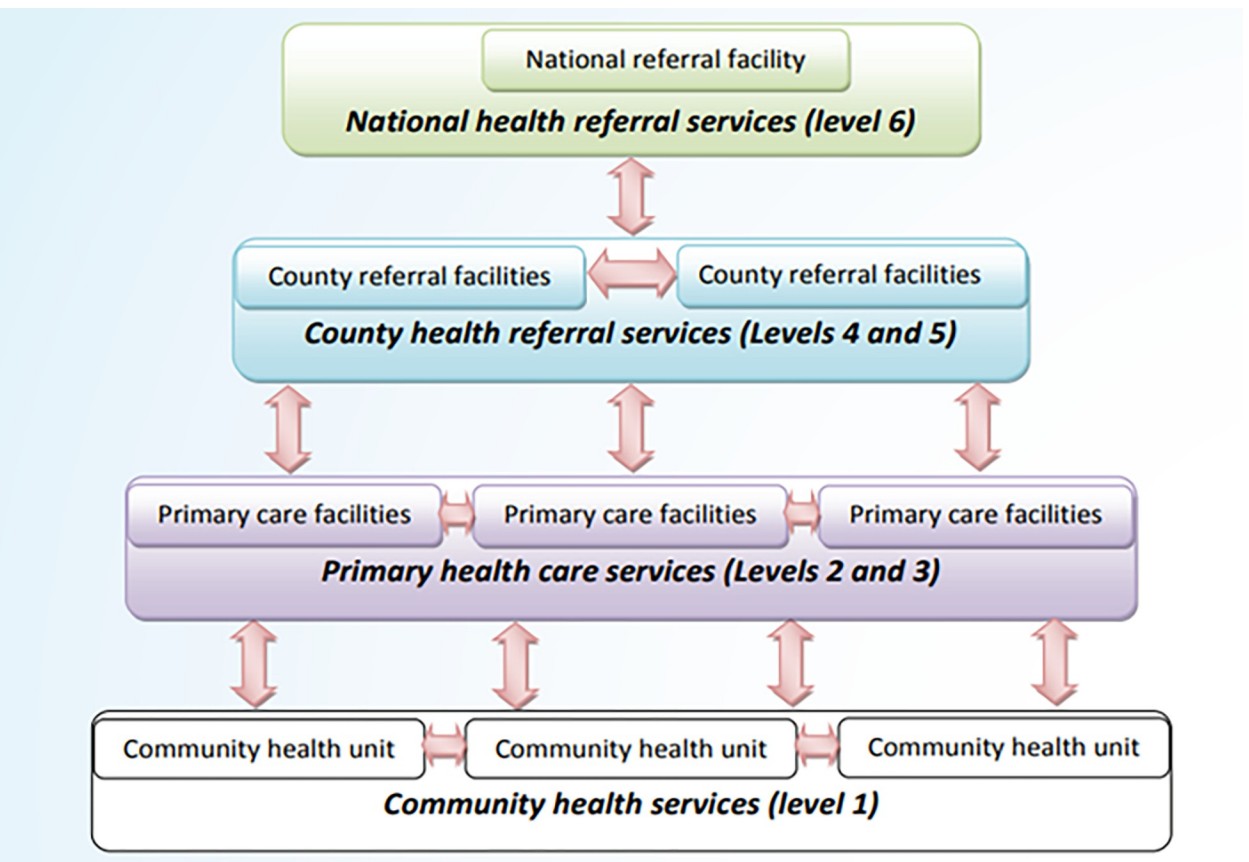

**Fig 1. Relationships between different levels of care (Reproduced from Kenya Health Sector Implementation Referral Guidelines, 2014).**

nurse-patient ratio of 1:10, which compromises not only the quality of nursing care given to patients but also the ability of KNH to effectively perform their statutory obligations [11–13].

At the strategic level, the government's overall strategic plan, the Second Medium-Term Plan (MTP) of Vision 2030, identifies the enforcement of the national referral guidelines as a key strategy to increase the use of services at lower levels of the health care system and reduce self-referral to higher levels of care. Consequently, on 1st July 2021, KNH management decided to reinforce the Kenya Health Sector Referral Guidelines 2014. This meant that patients would be seen based on referral letters from other health facilities to reduce the number of walk-in patients who would have otherwise been appropriately seen at the peripheral health facilities. This approach was anticipated to contribute to the decrease in KNH Orthopaedic admissions. No studies have been done to assess the effect of enforcement of national referral guidelines on orthopaedic admission patterns in sub-Saharan Africa, Kenya included. The purpose of this study was to determine the effect of the national referral guideline enforcement on the patterns of orthopaedic admissions to KNH.

## Methods and materials

### Study design

This was a pre-post intervention study. The pre intervention covered 5 months before while post intervention covered 5 months, after the enforcement of the national referral guidelines.

The variables namely socio-demographic characteristics (age, sex, marital status, religion, occupation, education level, smoking status and alcohol intake status), nature of admission (walk-ins vs facility referrals), occupation, point of admission (A&E, OC, COC), type of admission (emergency vs electives), and mode of payment (cash vs insurance) were compared before and after enforcement of the national referral guidelines.

The enforcement of the national referral guidelines required the referring health facility consult with the KNH referral Office for concurrence before patients are referred to KNH and that patients be accompanied with written official referral letters. This approach was used to ensure that only patients who required specialized medical and orthopaedic and trauma care unavailable at peripheral health facilities were admitted to KNH.

## Study setting

The KNH is the largest teaching and referral hospital in East and Central Africa and is based in Nairobi, the capital city of Kenya. It is located approximately 5km from the city centre. It has a bed capacity of 1,800, over 6,000 staff members, 50 wards, 22 outpatient clinics, 24 theaters (16 specialized) and an Accident & Emergency Department [3]. KNH Orthopaedic Wards were the study area. Of the 1800 bed capacity, 96 beds were allocated to orthopaedic wards. The KNH is a 10-floor storied building complex and the Orthopaedic wards are located on the 6th floor; however, we also have orthopaedic admissions on the 9th and 10th floors of private wings. Orthopaedic patients with other comorbidities were also admitted to other wards in the KNH. KNH receives complex and difficult to manage patients across the country and beyond. In addition to general orthopaedics, several speciality surgical services, including obstetrics and gynaecology, urology, cardiothoracic and vascular, plastic and reconstruction, neurosurgery, paediatric and general surgery, ear, nose, throat and maxillofacial surgery, are available in KNH.

## Study population

Orthopaedic inpatient caseloads.

## Study duration

The study duration was from February 1st to December 31st 2021. The study covered 5 months before (February 1, to June 30, 2021) and 5 months after (August 1, to December 31, 2021) enforcement of the national referral guidelines. The inclusion criterion was all orthopaedic admissions to KNH during the study period. The data collection period was from 1st January to 31st March 2022.

## Sample size calculation and sampling procedure

Using an adjusted Casagrande formula for calculating sample sizes that compare two binomial distributions [14], a total of 459 files and 446 files were sampled before and after the enforcement of the national referral guidelines. A sstratified sampling technique was used. Orthopaedic admissions were stratified by month of admission and Probability Proportional to Size (PPS) was used to determine the sample size per stratum. Systematic random sampling was then used to identify the sampled files using the KNH Health Records Register as the sampling frame from each of these three (3) service entry points {Accident and Emergency (A&E), Orthopedic Clinic (OC), and Corporate Outpatient Clinic (COC)}. This ensured that the selected sample was representative of the admissions by month from each of these three (3) orthopedic admission entry points.

## Variables

Variables abstracted for this study were: a) Socio-demographic characteristics–age, sex, marital status, occupation, religion, education level, smoking status and alcohol intake status b) nature of admission (walk-ins vs facility referrals); c) point of admission (A&E, OC, COC) d) type of admission (emergency vs electives); e) mode of payment (cash vs insurance).

## Data collection procedures

Data collection was done using a data abstraction form (S1 Text). This was a manual survey tool. Data were abstracted from the patient charts and transferred to the data abstraction form by the research assistants.

## Quality assurance and quality control procedures

Three (3) research assistants underwent a two (2) days training on the study protocol. A pilot study was conducted during the design of the study protocol to test the relevance and appropriateness of the data collection tools to answer the research questions and adjustments on data collection tools made as necessary. The Principal Investigator (PI) conducted daily reviews of all abstracted forms, verified for accuracy, completeness, and compliance with the research protocol. Inaccurate, incomplete and noncompliant entries were repeated under the supervision of the PI as part of ongoing training to ensure that similar errors were avoided.

## Ethical considerations

The University of Nairobi/Kenyatta National Hospital (UoN/KNH) Ethics and Research Committee granted ethical approval for the study (ERC No: P852/10/2021). Administrative approval was granted by the KNH Medical Research Department and KNH Orthopaedics Department. This study was carried out in accordance with UoN/KNH Ethics and Research Committee guidelines and regulations.

## Data management and analysis

The data abstraction tool was used to collect quantitative and qualitative data (S1 Text). The data were entered into a password-protected Redcap database and later exported to SPSS version 27.0 dataset for analysis (S1 File). Descriptive analysis was done by calculating frequency distribution for categorical variables. Socio-demographic characteristics namely age, sex, marital status, occupation, religion, education level, smoking status and alcohol intake status were compared pre and post intervention. Inferential analysis was done using Pearson's chi-square test and logistic regression to determine the statistical significance and strength of the associations. Pearson's chi-square test was used to compare nature of admission, point of admission, type of admission and mode of payment pre and post intervention. For regression analysis the outcome variable was period of study, that is, pre and post intervention study periods.

# Results

## Socio-demographic characteristics of the sample population

Overall, 905 charts were extracted, 459 were extracted before and 446 were extracted after the national referral guidelines enforcement.

The majority of patients were aged 25 to 64 years with children and elderly individuals (above 65 years) representing the least admissions. Over three quarters (75%) of the

**Table 1. Socio-demographic characteristics of orthopaedic admissions at KNH before and after enforcement of the national referral guidelines, 2021.**

| Variable | Category | Before (n = 459) | After (n = 446) | p-value |
|---|---|---|---|---|
| **Age** | 0–14 years | 59 (59.6%) | 40 (40.4%) | p = 0.068 |
| | 15–24 years | 83 (50.0%) | 83 (50.0%) | |
| | 25–64 years | 303 (50.5%) | 297 (49.5%) | |
| | Above 65 years | 14 (35.0%) | 26 (65.0%) | |
| **Sex** | Female | 83 (41.9%) | 115 (58.1%) | **(p = 0.005** |
| | Male | 374 (53.2%) | 329 (46.8%) | |
| | Missing | 2 | 2 | |
| **Marital status** | Married | 218 (48.9%) | 228 (51.1%) | p = 0.349 |
| | Minor | 61 (55.5%) | 49 (44.5%) | |
| | Separated & divorced | 20 (40.8%) | 29 (59.2%) | |
| | Single | 147 (53.5%) | 128 (46.5%) | |
| | Widow | 13 (54.2%) | 11 (45.8%) | |
| | Missing | | 1 | |
| **Religion** | Atheist | 2 (66.7%) | 1 (33.3%) | p = 0.609 |
| | Christian | 433 (50.1%) | 432 (49.9%) | |
| | Hindu | 1 (33.3%) | 2 (66.7%) | |
| | Muslim | 15 (62.5%) | 9 (37.5%) | |
| | Missing | 8 | 2 | |
| **Occupation** | Businessman/woman | 60 (52.6%) | 54 (47.4%) | p = 0.841 |
| | Casual | 204 (50.6%) | 199 (49.4%) | |
| | Employed | 64 (47.4%) | 71 (52.6%) | |
| | Other | 33 (55.9%) | 26 (44.1%) | |
| | unemployed | 92 (50.3%) | 91 (49.7%) | |
| | Missing | 6 | 5 | |
| **Education** | None | 33 (60.0%) | 22 (40.0%) | ***P = 0.001*** |
| | Preschool | 18 (81.8%) | 4 (18.2%) | |
| | Primary | 157 (51.0%) | 151 (49.0%) | |
| | Secondary | 166 (52.4%) | 151 (47.6%) | |
| | Tertiary | 74 (40.7%) | 108 (59.3%) | |
| | Missing | 11 | 10 | |
| **Smoking** | No | 348 (51.2%) | 332 (48.8%) | p = 0.780 |
| | Yes | 89 (50.0%) | 89 (50.0%) | |
| | Missing | 22 | 25 | |
| **Alcohol** | No | 258 (50.1%) | 257 (49.9%) | p = 0.522 |
| | Yes | 180 (52.3%) | 164 (47.7%) | |
| | Missing | 21 | 25 | |

The table shows the independent categorical variables extracted from the patient charts before and after enforcement of referral guidelines. Missing values ranging from 1 to 25 and were excluded from the analysis. The statistical analysis was performed by Pearson $\chi^2$ statistics at 95% CI using SPSS version 27.0.

admissions were male before and after the enforcement of the national referral guidelines. The male admissions peaked at 25–64 years of age and declined steadily to 65 years of age. Orthopaedic admissions were comparable across sexes for those older than 65 years. A good proportion of the admissions had primary or secondary education (Table 1).

Education and sex were significantly different between the two groups (p = 0.001 and p = 0.005, respectively). However, age, religion, occupation, marital status, smoking status, and alcohol intake habits were comparable between the two groups (P > 0.05) (Table 1).

**Table 2. Orthopaedic admissions before and after enforcement of the national referral guidelines, 2021.**

| Variables | | Before (n = 459) | After (n = 446) | Total (n = 905) | p-value | Logistic regression (p-value) |
|---|---|---|---|---|---|---|
| **Type of admission** | Walk-ins | 248 (54.9%) | 204 (45.1%) | 452 (49.9%) | *0.013* | 1.0 (REF) |
| | Facility referrals | 211 (46.6%) | 242 (53.4%) | 453 (50.1%) | | *1.653 (95% CI: 1.244–2.197) (p = 0.001)* |
| **Point of admissions** | Accident & Emergency (A&E) | 386 (84.1%) | 321 (72.0%) | 707 (78.1%) | *0.0001* | 1.0 (REF) |
| | Corporate Outpatient Clinic (COC) | 48 (10.5%) | 87 (19.5%) | 135 (14.9%) | *0.011* | *0.456 (95% CI: 0.311–0.668) (p <0.001)* |
| | Orthopaedic Clinics (OC) | 25 (5.4%) | 38 (8.5%) | 63 (7.0%) | 0.315 | 0.839 (95% CI: 0.453–1.552) (p = 0.575 |
| **Nature of admissions** | Elective | 70 (15.3%) | 118 (26.6%) | 188 (20.8%) | *0.0001* | 1.0 (REF) |
| | Emergency | 387 (84.7%) | 325 (73.4%) | 712 (78.7%) | | *0.498 (0.358–0.693)* |
| | Missing* | 2 | 3 | 5 (0.0%) | | |

The pearsons' $\chi^2$ statistical test was used to test for statistical significance at the 95% CI, and odds ratio estimate was used to test the strength of the association

* Represents the missing values which were not included in the statistical analysis. 1.0 (REF) refers to the reference group. Outcome: pre-post intervention periods.

## Effect of the enforcement of the national referral guidelines on orthopaedic admissions at KNH

**Profile of admissions to the KNH.** The proportion of walk-ins declined from 248 (54.9%) to 204 (45.1%) after the enforcement of the national referral guidelines, while the percentage of facility referrals increased from 211 (46.6%) to 242 (53.4%) during the same period (p = 0.013) (Table 2). After controlling for possible confounders (sex and education level), the odds of walk-ins were 1.653 (95% CI: 1.244–2.197) compared to facility referrals before the enforcement of the national referral guidelines. The odds of walk-ins were significantly greater before the enforcement of the national referral guidelines, and they decreased after the enforcement of the national referral guidelines.

Although the majority of admissions were admitted through the Accident and Emergency (A&E) department, there was a decline from 386 (84.1%) to 321 (72.0%) after the enforcement of the national referral guidelines (*p = 0.0001*), with the least admissions through the Orthopaedic Clinics (OC). There was also an increase in admissions through Corporate Outpatient Care (COC) after enforcement of the national referral guidelines from 48 (10.5%) to 87 (19.5%) (*p = 0.011*) (Table 2). Overall, there was a significant decrease in admissions through A&E and an increase in admissions thorough the COC and OC after the enforcement of the referral guidelines.

After the national referral guidelines were enforced, the number of emergency admissions decreased from 387 (84.7%) to 325 (73.4%), while the number of elective cases increased from 70 (15.3%) to 118 (26.6%) (*p = 0.0001*) (Table 2). Elective admissions were 50.2% less likely to occur before than after enforcement of the national referral guidelines (Table 2).

The study revealed that the proportions of referrals with official written referral letters were 48.3% and 49.2% before and after the enforcement of the national referral guidelines, respectively (p = 0.860) (Table 3).

**Table 3. Facility referrals with referral letters before and after enforcement of the national referral guidelines, 2021.**

| | | Facility referrals | | p-value |
|---|---|---|---|---|
| | | Before (n = 211) | After (n = 242) | |
| **Have referral letters** | No | 109 (51.7%) | 123 (50.8%) | p = 0.860 |
| | Yes | 102 (48.3%) | 119 (49.2%) | |

The Pearson $\chi^2$ statistical test was used to determine the statistical significance of the differences at 95% CI.

**Table 4. Multivariable analysis of key socio-demographic characteristics of orthopaedic admissions to KNH before and after enforcement of the national referral guidelines, 2021.**

| Characteristics | | Before (n = 459) | After (n = 446) | p-value | Logistic regression (95% CI) |
|---|---|---|---|---|---|
| **Sex** | Female | 83 (18.2%) | 115 (25.9%) | p = 0.005 | 1.0 (REF) |
| | Male | 374 (81.8%) | 329 (74.1%) | | *1.575 (1.145–2.166)* |
| | Missing* | 2 | 2 | | |
| **Education level** | None | 33 (7.4%) | 22 (5.0%) | p = 0.001 | 1.0(REF) |
| | Preschool | 18 (4.0%) | 4 (0.9%) | | 3.0 (0.894–10.063) |
| | Primary | 157 (35.0%) | 151(34.6%) | | 0.693 (0.387–1.243) |
| | Secondary | 166 (37.1%) | 151 (34.6%) | | 0.733 (0.409–1.313) |
| | Tertiary | 74 (16.5%) | 108 (24.8%) | | *0.457 (0.247–0.845)* |
| | Missing* | 11 | 10 | | |
| **Mode of payment** | Cash | 344 (75.3%) | 277 (62.2%) | p = 0.0001 | 1.0(REF) |
| | Insurance | 113 (24.7%) | 168 (37.8%) | | *0.538 (0.404–0.716)* |
| | Missing* | 2 | 1 | | |

The pearson $\chi^2$ tests was used to determine the statistical significance of the differences and odds ratio (95% CI) was used to test the strength of the associations. 1.0 (REF) refers to reference group

* Represents missing values and were not included in the statistical analysis.

**Key socio-demographic patterns of admissions to KNH.** The study revealed that the likelihood of being male was 1.575 (1.145–2.166) less likely than that of being female after the enforcement of the national referral guidelines (*p = 0.005*) (Table 4).

With regard to education level, the study demonstrated that the likelihood of having a tertiary level of education was 54.4% greater after the enforcement of the national referral guidelines (*p = 0.001*) (Table 4).

Regarding the mode of payment, the odds of being a cash payer were 1.846 (1.387–2.458) greater before the enforcement of the national referral guidelines while the likelihood of having insurance cover was 46.2% less likely before the enforcement of the national referral guidelines (Table 4).

The mode of payment and its association with sex were reviewed, and there was no statistically significant association before the enforcement of the national referral guidelines. However, after the enforcement of the referral guidelines, males were 2.366 times more likely to be cash payers than females (Table 5).

**Association between patients' characteristics and the nature of admissions.** The study demonstrated that among walk-ins, those aged 15–24 years and 25–64 years had a statistically

**Table 5. Multivariable analysis of sex differences in orthopaedic admissions to KNH before and after enforcement of the national referral guidelines, 2021 disaggregated by mode of payment.**

| | Sex | Cash | Insurance | p-value | OR (95% CI) |
|---|---|---|---|---|---|
| **Before** | Female | 58 (17.0%) | 25 (22.1%) | p = 0.218 | 1.0 (REF) |
| | Male | 284 (83.0%) | 88 (77.9%) | | 1.391 (0.822–2.355) |
| **After** | Female | 54 (19.6%) | 61 (36.5%) | *p = 0.0001* | 1.0 (REF) |
| | Male | 222 (80.4%) | 106 (63.5%) | | *2.366 (1.534–3.649)* |

The Pearson $\chi^2$ test was used to determine the statistical significance of the differences and odds ratio (95% CI) was used to test the strength of the associations. 1.0 (REF) refers to the reference group.

**Table 6. Association between age and point of admission before and after enforcement of the national referral guidelines stratified by facility referral and walk-ins status, 2021.**

| | Age category | Point of admission | Before (n = 459) | After (n = 446) | p-value |
|---|---|---|---|---|---|
| **Walk-ins** | 0–14 years | A&E | 13 (72.2%) | 5 (27.8%) | p = 0.546 |
| | | OC | 6 (54.5%) | 5 (45.5%) | |
| | | COC | 5 (55.6%) | 4 (44.4%) | |
| | 15–24 years | A&E | 49 (69.0%) | 22 (31.0%) | *p = 0.0001* |
| | | OC | 0 (0.0%) | 6 (100.0%) | |
| | | COC | 2 (20.0%) | 8 (80.0%) | |
| | 25–64 years | A&E | 129 (62.3%) | 78 (37.7%) | *p = 0.001* |
| | | OC | 11 (39.3%) | 17 (60.7%) | |
| | | COC | 28 (40.0%) | 42 (60.0%) | |
| | Above 65 years | A&E | 3 (30.0%) | 7 (70.0%) | p = 0.628 |
| | | OC | 0 (0.0%) | 2 (100.0%) | |
| | | COC | 2 (20.0%) | 8 (80.0%) | |
| **Facility referral** | 0–14 years | A&E | 30 (55.6%) | 24 (44.4%) | p = 0.709 |
| | | OC | 3 (75.0%) | 1 (25.0%) | |
| | | COC | 2 (66.7%) | 1 (33.3%) | |
| | 15–24 years | A&E | 31 (40.8%) | 45 (59.2%) | p = 0.243 |
| | | OC | 1 (100.0%) | 0 (0.0%) | |
| | | COC | 0 (0.0%) | 2 (100.0%) | |
| | 25–64 years | A&E | 123 (48.2%) | 132 (51.8%) | p = 0.075 |
| | | OC | 4 (40.0%) | 6 (60.0%) | |
| | | COC | 8 (26.7%) | 22 (73.3%) | |
| | Above 65 years | A&E | 8 (50.0%) | 8 (50.0%) | p = 0.368 |
| | | OC | 0 (0.0%) | 1 (100.0%) | |
| | | COC | 1 (100.0%) | 0 (0.0%) | |

The pearson $\chi^2$ test was used to determine the statistical significance of the differences at 95% CI.

significant decline in admissions through the A&E after the enforcement of the national referral guidelines (Table 6).

The study revealed a statistically significant increase in the proportion of non-trauma-related orthopaedic admissions after the national referral guidelines were enforced among walk-in admissions ($p < 0.001$). The odds of non-trauma-related admissions were 2.829 times greater after the enforcement of the national referral guidelines among walk-ins' admissions (Table 7).

**Table 7. Association between the nature of injury before and after the enforcement of the national referral guidelines stratified by facility referral and walk-ins, 2021.**

| | Nature of Injury | Before | After | p-value | OR |
|---|---|---|---|---|---|
| **Walk-in** | Trauma related | 204 (62.0%) | 125 (38.0%) | *p p<0.001* | *2.829 (1.842–4.345)* |
| | Non–Trauma related | 45 (36.6%) | 78 (63.4%) | | |
| **Facility Referral** | Trauma related | 200 (47.6%) | 220 (52.4%) | p = 0.148 | 1.736 (0.816–3.690) |
| | Non–Trauma related | 11 (34.4%) | 21 (65.6%) | | |

The Pearson $\chi^2$ test was used to determine the statistical significance of the differences and odds ratio (95% CI) was used to test the strength of the associations.

## Discussion

### Socio-demographic characteristics of the sample population

The study demonstrated that more than over three-quarters of the orthopaedic admissions were male before and after the enforcement of the national referral guidelines during the study period. This finding compares favourably with studies done in Addis Ababa, Ethiopia, India, Nepal, the Middle East, and Botswana that showed that males predominated in orthopaedic admissions [15–23]. In addition, male orthopaedic admissions predominated and peaked at 25–64 years of age; these admissions declined steadily to 65 years of age while female admissions which were mostly elective cases, predominated. This finding compares favourably with a study in Botswana that showed above 60 years of age, females were disproportionately affected [22]. Orthopaedic admissions were comparable across sexes for those older than 65 years. This finding is comparable to that of a retrospective study done in a tertiary hospital in Nepal which showed similar admission rates in patients older than 60 years [18].

Among the age groups, children and those aged above 65 years had the least orthopaedic admissions both before and after the enforcement of the national referral guidelines, while the majority of admissions were observed among those aged group 25–64 years. This compares with a retrospective study on orthopaedic admissions done in Uganda, Nigeria, Botswana, South Africa, Warangal and Delhi, India, Taiwan, Brazil, the United States of America and England that showed that the majority of admissions were young [15, 24–35]. There was a bimodal distribution with high rates of admission for young adults up to the age of 35 years as well as for those above 45 years old which is comparable to the findings of studies done in India [15]. Similar studies in Tanzania and Taiwan also depicted low paediatric orthopaedic admissions [27, 35, 36]. This finding contrasts with that of a study done in PCEA Kikuyu Mission Hospital, Kenya, in which 18.84% orthopaedic admissions were of paediatric age [23]. This could be because the PCEA Kikuyu Mission Hospital is an established, faith-based hospital and highly regarded specialized orthopaedic centre in the country. Faith based mission hospitals provide majority of surgical care outside the public referral hospitals in Kenya [37, 38].

However, the study revealed a relative increase in female orthopaedic admissions compared to males after the enforcement of the national referral guidelines. This may be attributed to a reduction in the number of emergency admissions through the Accident and Emergency department since the majority of these emergency admissions were male. This finding is consistent with those of a study done in Rift Valley Provincial Hospital (currently Nakuru County Referral Hospital), Kenya, Uganda, South Africa, Taiwan which showed that the majority of emergency admissions through road traffic accidents and interpersonal violence were males [27, 32, 34, 39].

The enforcement of the national referral guidelines was associated with more orthopedic admissions having a tertiary level of education, non-trauma-related conditions and admissions with active insurance cover. These may have been driven mainly by admissions through the COC (the COC is the private clinic wing of the KNH) following the enforcement of the national referral guidelines.

### Effect of the referral guidelines on orthopaedic admissions at KNH

On 1$^{st}$ July 2021, the KNH enforced the national referral guidelines that were meant to streamline the referral process from peripheral health facilities and allow the KNH to a) manage complex orthopaedic cases that cannot be handled at lower-level health facilities as per the Kenya Health Sector Referral Implementation Guidelines, 2014. This resulted in a statistically significant reduction in walk-ins and an increase in facility referrals. Patients were encouraged to

seek services from lower-level health facilities and they were also encouraged to come to KNH with a written formal referral letter from the referring health facility.

There was a significant decrease in admissions through A&E and an increase in admissions through the COC and OC after the enforcement of the national referral guidelines. This means that the enforcement of the national referral guidelines was associated with an increase in admissions through clinics as opposed to A&E. This is in tandem with reductions in the proportions of emergency admissions from 84.7% to 73.4% and a relative increase in the proportions of elective admissions after enforcement of referral guidelines. In other words, enforcement of the national referral guidelines was associated with more elective orthopaedic admissions and fewer emergency orthopaedic admissions, especially among walk-ins aged 15–64 years. The majority of these emergency admissions through the A&E department were young and middle-aged admissions, for which a significant decline was reported after the enforcement of the national referral guidelines. Despite the decline in A&E admissions, A&E orthopaedic admissions still represent the major points of admission to KNH, which is comparable to findings of a retrospective study on traumatic injury admissions at Kilimanjaro Christian Medical Centre in Tanzania, a referral health facility in northwestern Tanzania, a tertiary health facility in Nigeria that also showed that the majority of admissions were through the emergency department [40–42]. In this study, the decreases in emergency admissions reflect the fact that these patients may have been handled at lower-level facilities as opposed to being referred to KNH post enforcement of national referral guidelines.

In addition, there was a statistically significant increase in the percentage of non-trauma orthopaedic admissions doubling from 12.0% to 22.4% after enforcement of the national referral guidelines, which also reinforced the fact that elective orthopaedic admissions increased while emergency admissions decreased after the enforcement of the national referral guidelines. Non-trauma orthopaedic cases are usually elective admissions. This finding compares favourably with that of a study done in Moshi, Tanzania in which 20% of admissions were non-trauma-related [43]. However, these study findings differ from those of a retrospective study in India in which non-trauma orthopaedic admissions accounted for one-third of all orthopaedic admissions [15]. This means that non-trauma orthopaedic admissions more often than not represent minority orthopaedic admissions.

The enforcement of the national referral guidelines was associated with a statistically significant increase in the proportion of orthopaedic admissions with active insurance cover among walk-ins' admissions. The increase in the proportion of patients with insurance cover to one-third of the admissions after the enforcement of the national referral guidelines could be due to an increase in the number and proportion of female orthopedic admissions through the clinics. This finding is in agreement with a study done in Moi Teaching and Referral Hospital (MTRH) in Eldoret, Kenya, which showed that women were more likely to arrive at general surgery through the clinic and were also more likely than men to have insurance cover [44]. This, however, differs with a study done in Tenwek Mission Hospital on surgical gastrointestinal disorders that showed less women accessing care [45].The increase in elective cases was driven mostly by elderly women with degenerative musculoskeletal disorders who had a tertiary level of education and active insurance cover. These patients were admitted mostly through the clinics. This reinforced the finding that the proportion of non-trauma-related admissions increased after enforcement of the national referral guidelines. The study showed insurance coverage of one-third of admissions post-enforcement of national referral guidelines, which differs from the findings of the study done in MTRH on General Surgery admissions, which showed that insurance cover stood at one- half of admissions [44]. This is partly because MTRH are proactive in registering elective surgical patients to the National Health Insurance Fund prior to scheduling surgery to defray anticipated personal costs. However, this

study focused on orthopedic conditions which were mostly emergency admissions and that could explain the lower insurance coverage in the current study. Nonetheless, this insurance coverage did increase after the enforcement of the national referral guidelines due to relative increase in female elective admissions in patients with non-trauma-related conditions. This study contradicts a study done in Rwanda that revealed that 82.2% had community health insurance, 5.7% had other private insurance and only 12.1% were not insured [46]. The difference is because in Rwanda the government facilities under review were funded by a Boston-based non-governmental organization that partnered with the government of Rwanda to improve health care delivery.

However, there was no significant difference in facility referrals being accompanied by official written referral letters from the referring facilities to the KNH. The study showed low use of referral letters. One-half of the facility referrals had written referral letters. This finding contradicts the findings of studies done at Moi Teaching and Referral Hospital, Kenya, Nigeria and Uganda which showed that more than two-thirds of facility referrals had referral slips from the referring health facilities [47–49]. The comparatively low use of referral letters in the current study could be because most of the referrals were verbal about the telephone and once a verbal consensus had been reached, the referring health facilities did not see the need to write a formal written referral letter. This finding also contrasts with a study on adherence to referral guidelines and use of referral letters in Cape Town, South Africa, Malawi and the United States of America which revealed that the vast majority of patients adhered to referral guidelines with the use of formal referral letters [33, 50–52]. This is likely because these health facilities were highly specialized national and regional treatment centres.

This study has a few limitations. First, there were incomplete or missing records because the data were collected retrospectively. This was mitigated by increasing the sample size by 10%. Second, the COVID-19 pandemic may have affected the referrals of patients from peripheral health facilities and walk-in patients. This was addressed by ensuring that the data collection period covered the COVID_19 period when the restrictions of inter-county movements were lifted. Third, this study is retrospective and a quasi-experimental in nature and therefore weak in determining causality. Fourth, the single hospital nature of the study limits generalizability of the findings despite its huge catchment population. Despite these limitations, given the paucity of published literature on this study topic, this study offers key information on the effects of the enforcement of referral guidelines in Orthopaedic admissions in KNH with important lessons for Kenya and possibly sub-Saharan Africa.

## Conclusions

In conclusion, the enforcement of the national referral guidelines significantly reduced the proportion of walk-in patients from 54.9% to 45.1%, while facility referrals increased from 46.6% to 53.4%. Efforts should be made to ensure the gains made are sustained through strict enforcement of the national referral guidelines through development and dissemination of the referral standard operating procedures, supervisions, referral forms. There was also an increase in orthopaedic admissions through the clinics after the national referral guidelines were enforced. The proportion of emergency admissions declined while that of elective admissions increased after the enforcement of the national referral guidelines. The increase in elective cases was mainly driven by the increase in female admissions with active insurance cover, tertiary education, non-trauma related conditions and older age groups. While the enforcement of the national referral guidelines had no effect on the use of official written referral letters, it did limit access and utilization of orthopaedic services in KNH by younger male patients with no active insurance, lower education status and who required emergency orthopedic care.

Consequently, it created disparities in access to orthopedic care at KNH. Further studies should be done to better understand the fate of these young males with no active insurance cover and with emergency orthopedic conditions with a view to address the emerging disparities to access to orthopedic care in KNH imposed by enforcement of the national referral guidelines.

## Supporting information

**S1 File. Working.** 05.06.2022.
(SAV)

**S1 Text. Data abstraction tool.**
(DOCX)

## Acknowledgments

We would like to sincerely acknowledge the work of Brian Okinyi and Micah J. Kipkemei for their commitment and assistance in the data collection process.

## Author Contributions

**Conceptualization:** Maxwell Philip Omondi, Joseph Mwangi Chege, Herbert Ong'ang'o, Fred Chuma Sitati.

**Data curation:** Maxwell Philip Omondi, Joseph Mwangi Chege.

**Formal analysis:** Maxwell Philip Omondi.

**Funding acquisition:** Maxwell Philip Omondi, Joseph Mwangi Chege.

**Investigation:** Maxwell Philip Omondi, Fred Chuma Sitati.

**Methodology:** Maxwell Philip Omondi, Fred Chuma Sitati.

**Project administration:** Maxwell Philip Omondi, Fred Chuma Sitati.

**Resources:** Maxwell Philip Omondi.

**Software:** Maxwell Philip Omondi.

**Supervision:** Maxwell Philip Omondi, Joseph Mwangi Chege, Herbert Ong'ang'o, Fred Chuma Sitati.

**Validation:** Maxwell Philip Omondi, Joseph Mwangi Chege, Herbert Ong'ang'o, Fred Chuma Sitati.

**Visualization:** Maxwell Philip Omondi, Fred Chuma Sitati.

**Writing – original draft:** Maxwell Philip Omondi, Fred Chuma Sitati.

**Writing – review & editing:** Maxwell Philip Omondi, Fred Chuma Sitati.

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
