## [Decision Letter · Decision Letter 0]

21 Nov 2023

PONE-D-23-24470Effect of enforcement of referral guidelines on patterns of orthopedic and trauma admissions to Kenyatta National Hospital, Kenya: pre-posttest study designPLOS ONE

Dear Dr. Omondi,

Thank you for submitting your manuscript to PLOS ONE. After careful consideration, we feel that it has merit but does not fully meet PLOS ONE’s publication criteria as it currently stands. Therefore, we invite you to submit a revised version of the manuscript that addresses the points raised during the review process.

We look forward to receiving your revised manuscript.

Kind regards,

Dickens Otieno Onyango

Academic Editor

PLOS ONE

Journal Requirements:

"This study was partially funded by Kenyatta National Hospital (https://knh.or.ke/). MPO received the funding award. The research grant award was RFA 2020/21."

4. We are unable to open your Supporting Information file WORKING. 05.06.2022.sav. Please kindly revise as necessary and re-upload.

Reviewers' comments:

Reviewer's Responses to Questions

**Comments to the Author**

1. Is the manuscript technically sound, and do the data support the conclusions?

Reviewer #1: Partly

Reviewer #2: Yes

2. Has the statistical analysis been performed appropriately and rigorously? 

Reviewer #1: No

Reviewer #2: Yes

3. Have the authors made all data underlying the findings in their manuscript fully available?

Reviewer #1: Yes

Reviewer #2: Yes

4. Is the manuscript presented in an intelligible fashion and written in standard English?

Reviewer #1: No

Reviewer #2: Yes

5. Review Comments to the Author

Reviewer #1: Review

Line 69-72 – The author states that tertiary health care is compromised by the large demand for primary health care and in the process cripples the primary healthcare system and ensure that the PHC facilities remain underutilized and inefficient.

This paragraph is contradictory and the author needs to restate the message for clarity.

Line 75 – Nairobi City County is the correct name of ‘Nairobi County’

Line 74-83: could do with revision so that there is less repetition and clarity in the message

Line 101-107: The author nearly concludes that the high bed occupancy is the single reason responsible for the low nurse-patient ratio. This may not be necessarily accurate, causality is not clearly demonstrated hence the need to reword this so that it is one of the possible reasons

Line 114-121 – Is there any data on self referral for orthopedic cases, what is the acceptable self-referral rate? And what are the dangers to the health system of this? I would have liked to hear just a little bit more on this in the Introduction section.

Overall comments:

The introduction section needs to be reworked a bit to tell the story of the author. As it is, it states facts but they do not flow very well into each other to tell the story.

Methods and materials

Line 125 – ‘….concurrence before that patients are referred to KNH.’ This has a grammatical error

Line 125-127: ‘This is to ensure only a) patients who require specialized medical and orthopaedic and trauma 126 care or b) no neighbouring government health facility has the capacity to manage the cases, get admitted to KNH.’ This requires rewriting to avoid grammatical errors and make it clear

Study area

Line 128-131: Needs more detail, how many floors does the hospital sit on? Staff establishment? Of the 2000 beds, how many are allocated to the orthopaedic ward? This would be a good section to introduce the orthopaedic wards in detail: talk about the bed capacity and the staffing issues and any other contextual information that would make the reader understand the state

Line 133-140: Need to be reworded to be more concise. There is a bit of repetition especially on the date of guideline enforcement and . Once you state the date and the review duration, you will not need to granulate it further like you have.

Line 156-157: Please describe the data collection tool more, was it an electronic or manual one? How were data handled if it was manual?

Line 158-164: What happened in the event of inaccuracy, incompleteness or non-compliance?

Line 178-181: section (b) is missing. The effect of COVID is not clearly stated and the mitigation measures not understood. COVID probably decreased the number of referrals due to many reasons (restrictions in movement, change of SOP in hospitals etc.). Additionally, it would be good to contextualize the current state of the pandemic at the time of data collection since there were several phases characterized by variation sin regulations at every phase. i.e. at some point, restrictions were lifted, referrals were allowed etc. So at what point were we during the data collection.

Results Section

The first statement should include the total number of cases included in the study. This is missing.

Line 183-189: The text goes straight into multivariable analysis and we do not get to hear a little more description of the various socio-demographic variables in the text.

Line 198: Proportion of walk ins – what is the denominator? This needs to be stated here

Line 203-205: This requires to be re-interpreted and stated. As is, it is not very clear

Line 238-245: Part of this paragraph sounds like a justification for the study or a discussion. May be best to simply report the results for this section and move the other descriptions to the their more appropriate sections

Line 249: The Legend seems to have become part of the paragraph (and I suspect that this was not the intention)

Line 258-262: The interpretation appears to be incorrect

Line 306: Is that really true? That because ‘The odds of trauma-related admissions were 2.829 times higher before enforcement of referral guidelines compared to after enforcement of referral guidelines amongst walk-ins’ admissions’ then ‘This means that there was a statistically significant increase in non-trauma-related admissions after enforcement of the referral guidelines amongst walk-ins. ’. I doubt that this statements are the mirror opposite of each other. There are probably confounders affecting this result.

Line 319-324: these are the sentences I expected to see in the results’ section i.i an account of the findings and then in the discussion we would have discussed these findings, compared and contrasted them with similar studies. Otherwise this is like a new introduction of data that has not been previously reported elsewhere

Line 336: the probable major reason is that PCEA Kikuyu is a specialized centre and not really that it is a private mission hospital

Line 338-343: That disparity in the number/proportion of female admissions cannot be quite explained by the enforcement of the guidelines. Let’s dig deeper for some feasible reasons for this.

Line 345-348: An association is bipartite between a dependent and independent variable. Your discussion tends towards a tripartite description i.e. enforcement of gudielines, orthopedic trauma and a 3rd factor etc. This needs to be refined so it is clear which factors are being evaluated against which variable

Line 355-357: needs to be reworded to be a bit ‘tighter’, esp line 356.

Line 360: thorough should be through

Line 361-363: we need to be clear on the dependent vs independent variable

Line 372-376: these appear misplaced and do not quite communicate

Line 378-387: This paragraph has 2 themes: comparing the association in the increase in trauma admissions with insurance with enforcement of guidelines. At Line 384, the other opines that this increase could be due to increase in the number and proportion of female cases. The conclusion that this is linked with the fact that these women had an active insurance cover may need a bit more of argument.

Line 389-393: Is there a result that communicates this consensus?

Recommendations

They need to have a specific individual or office actioned otherwise they are not quite measurable

Reviewer #2: The study topic is relevant and tries to address current concerns about referral procedures and pathways. The results are logical but the author should;

1. Remove un-important sections in the background. I found one particular paragraph not linking up

2. Adjust and give more information about the study area and the study population

3. Make result areas more clear. You can create small sub headings for easy flow

4. Only discuss significant findings.

5. Where possible, the author should try and give their interpretation of the trends.

5. There is need to find more literature to relate to or support the findings. Most of the findings in this study are not supported by other literature!

6. PLOS authors have the option to publish the peer review history of their article (what does this mean?). If published, this will include your full peer review and any attached files.

Reviewer #1: No

Reviewer #2: **Yes: **Oyugi O. Samuel,

---

## [Author Response · Author response to Decision Letter 0]

1 Dec 2023

We have addressed the reviewers comments to the best of our abilities.

There is paucity of published literature on this topic and so not much really exist for robust comparison and contrasts in the discussion section

---

## [Decision Letter · Decision Letter 1]

19 Dec 2023

PONE-D-23-24470R1Effect of enforcement of referral guidelines on patterns of orthopedic and trauma admissions to Kenyatta National Hospital, Kenya: pre-posttest study designPLOS ONE

Dear Dr. Omondi,

Thank you for submitting your manuscript to PLOS ONE. After careful consideration, we feel that it has merit but does not fully meet PLOS ONE’s publication criteria as it currently stands. Therefore, we invite you to submit a revised version of the manuscript that addresses the points raised during the review process.

**ACADEMIC EDITOR: ****Study design**: "This was a pre-post study design." This statement describing the study design is inadequate. Could the authors provide more details about the study design, such as what period was compared with what period and what was compared.**Study setting** (referred to as "study area" in the manuscript): While the authors describe the geographic location and physical infrastructure at KNH, crucial information about the current referral protocols in Kenya and KNH, vital for understanding the study context, is absent. To clarify the setting for the reader, please provide details on how patients navigate Kenya's healthcare system to reach KNH and the specific referral practices and guidelines in place at the hospital itself. Please also include information on how the refferal guidelines were enforced and when this enforcement began. **Limitations: **These have been included in the methods section. Please discuss your study limitations in the discussion section. **Results: This section includes information that belongs to the methods and discussion sections, such as "**Key socio-demographic characteristics of orthopaedic and trauma admissions at KNH such as age, sex, education, occupation, marital status, smoking and alcohol status were reviewed and compared between the two groups.""This means that there was a statistically significant increase in non-trauma-related admissions after enforcement of the referral guidelines amongst walk-ins."Revise this section thoroughly to remove all references to study methods or discussion. In this section only present your results. No methods, no discussion.Tables - please revise the formatting of all tables to make them visually appealing and less crowded. Several tables presenting stratified analyses (Table 2 to 5) each contain only one variable. It is not clear why the authors decided to present it this way. A preferable way would be to include all these variables in one table.==============================

We look forward to receiving your revised manuscript.

Kind regards,

Dickens Otieno Onyango

Academic Editor

PLOS ONE

Reviewers' comments:

Reviewer's Responses to Questions

**Comments to the Author**

1. If the authors have adequately addressed your comments raised in a previous round of review and you feel that this manuscript is now acceptable for publication, you may indicate that here to bypass the “Comments to the Author” section, enter your conflict of interest statement in the “Confidential to Editor” section, and submit your "Accept" recommendation.

Reviewer #1: All comments have been addressed

Reviewer #2: (No Response)

2. Is the manuscript technically sound, and do the data support the conclusions?

Reviewer #1: Yes

Reviewer #2: No

3. Has the statistical analysis been performed appropriately and rigorously? 

Reviewer #1: Yes

Reviewer #2: No

4. Have the authors made all data underlying the findings in their manuscript fully available?

Reviewer #1: Yes

Reviewer #2: Yes

5. Is the manuscript presented in an intelligible fashion and written in standard English?

Reviewer #1: Yes

Reviewer #2: No

6. Review Comments to the Author

Reviewer #1: The author has effected the revisions as requested with a clear description of the bed capacity and orthopaedic case load.

The results section begins with a summative response of the total cases studied.

The scientific language is also tightened.

The recommendations are now specific, and actioned to various entities

Reviewer #2: I reject the manuscript.

1. The manuscript is not up to standard for publication in this journal as it is. The revisions made are not adequate.

3. The author has not adequately supported the findings in this study. No much is done to relate the findings with previous related studies

4. There is need to refine the flow and grammer in the entire manuscript- the author to avoid romans and bullets, exhaustively discuss and the findings, make precise and specific recommendations.

7. PLOS authors have the option to publish the peer review history of their article (what does this mean?). If published, this will include your full peer review and any attached files.

Reviewer #1: No

Reviewer #2: **Yes: **OYUGI SAMUEL

---

## [Author Response · Author response to Decision Letter 1]

25 Dec 2023

We have responded to the new set of comments and uploaded the response to reviewer 2. 

We are still open for more reviews to improve the manuscript further. All comments raised so far have been addressed to the best of our ability.

Thanks and Merry Christmas and Happy New Year

---

## [Decision Letter · Decision Letter 2]

31 Jan 2024

PONE-D-23-24470R2Effect of enforcement of referral guidelines on patterns of orthopedic and trauma admissions to Kenyatta National Hospital, Kenya: pre-posttest study designPLOS ONE

Dear Dr. Omondi,

Thank you for submitting your manuscript to PLOS ONE. After careful consideration, we feel that it has merit but does not fully meet PLOS ONE’s publication criteria as it currently stands. Therefore, we invite you to submit a revised version of the manuscript that addresses the points raised during the review process.

We look forward to receiving your revised manuscript.

Kind regards,

Dickens Otieno Onyango

Academic Editor

PLOS ONE

Journal Requirements:

Reviewers' comments:

Reviewer's Responses to Questions

**Comments to the Author**

1. If the authors have adequately addressed your comments raised in a previous round of review and you feel that this manuscript is now acceptable for publication, you may indicate that here to bypass the “Comments to the Author” section, enter your conflict of interest statement in the “Confidential to Editor” section, and submit your "Accept" recommendation.

Reviewer #1: (No Response)

Reviewer #2: All comments have been addressed

2. Is the manuscript technically sound, and do the data support the conclusions?

Reviewer #1: Yes

Reviewer #2: Yes

3. Has the statistical analysis been performed appropriately and rigorously? 

Reviewer #1: Yes

Reviewer #2: Yes

4. Have the authors made all data underlying the findings in their manuscript fully available?

Reviewer #1: Yes

Reviewer #2: Yes

5. Is the manuscript presented in an intelligible fashion and written in standard English?

Reviewer #1: No

Reviewer #2: Yes

6. Review Comments to the Author

Reviewer #1: The author has done a commendable job to revise the manuscript and it now reads better. The various sections do not have misplaced information, the manuscript is nor summarized better and the recommendations provided are specific.

However we have an opportunity to improve. I provide the following comments and recommendations:

Line 112-113 – requires grammatical edits

Line 118-121 – move the detailed timelines to study duration section below (line 160) so that there is no repetition

Line 150 – needs to be edited to read ‘all patients being referred to KNH…’

Line 151-156 – is quite repetitive, could be summarized more concisely to communicate that 1) a decision was made, 2) it was enforced and 3) it was communicated (to whom and how)

Line 180 – from whom was informed consent sought since this was a retrospective study? How did we seek auditory and visual consent? This may need to be moved to the paragraph on ethical considerations

Line 273-274 –May be better to reword this or altogether move this line to the interpretation/discussion section.

Line 279 – delete the word ‘notable’ and state the result as is

Line 281 - delete the word ‘notable’ and state the result as is

Line 345 - delete the word ‘about’

Reviewer #2: The author has tried to address the issues raised and has related his finding to other previous researches despite the study area having limited literature.

7. PLOS authors have the option to publish the peer review history of their article (what does this mean?). If published, this will include your full peer review and any attached files.

Reviewer #1: No

Reviewer #2: **Yes: **Oyugi Samuel

---

## [Author Response · Author response to Decision Letter 2]

4 Feb 2024

I have reviewed and responded to the comments. I have also gone through the document and edited some typographical errors.

---

## [Decision Letter · Decision Letter 3]

16 Apr 2024

PONE-D-23-24470R3Effect of enforcement of referral guidelines on patterns of orthopedic and trauma admissions to Kenyatta National Hospital, Kenya: pre-posttest study designPLOS ONE

Dear Dr. Omondi,

Thank you for submitting your manuscript to PLOS ONE. After careful consideration, we feel that it has merit but does not fully meet PLOS ONE’s publication criteria as it currently stands. Therefore, we invite you to submit a revised version of the manuscript that addresses the points raised during the review process.

**ACADEMIC EDITOR:****1 - The manuscript has numerous grammatical errors. Thorough editing for langauage is needed. Below is an example:**** - "Kenyatta National Hospital (KNH) are managed by the national government ..."****- There are some long and winding statements that are not easy to follow. The authors could revise to use clear and concise statements. ****- line 96 to 99 is a very short paragraph that should be combined with the preceding paragraph****2 - Repetition - The concept of KNH being a national referral hospital is repeated at least four times in the introduction and several times in the methods section. The authors could revise to avoid repetition. This would make the article concise and easy to follow. ****3 - The introduction section does not identify a clear gap in the literature that this study is filling and does not clearly describe the objective of this study.****5 - Please clarify on the study design - currently the design is described as "preposttest". The authors proceed to mention pretest and post test.  Readers might interpret this as having administered a pretest followed by a post test evaluation as is the case when trainings are done. It might be better to describe this as a retrospective abstraction of medical records that were used for a pre- versus post-intervention evaluation. The pre versus post analysis is used to assess the effect of the intervention on some indicators of the effectiveness of a refferral system. ****6 - Study setting section is very long  and repeats concepts already discussed in the introduction section. The details provided here about KNH should be those that are relevant to this evaluation and should be focussed on orthopedics/trauma.****7 - The section headed " Data management and analysis plan"  - the word "plan" should be removed. This section should clearly explain how data was analyzed. "Descriptive analysis was done using frequency distribution" - probably the authors meant that descriptive analysis was done by calculating frequencies for categorical variables and measures of central tendency/dispersion for continous variables. The statements about inferential analysis and logistic regression are also vague. Please in this section be clear about how the pre- versus post-intervention analysis was conducted - what was compared with what and which statistical tests were used. For logistic regression - what was the outcome of interest? What covariates were used and how was the final model selected?****8 - Results - it is not clear from table 2 & 4 what the logistic regression results mean and what was the outcome of interest and what was the measure of association and whether the presented results are adjusted or not. **9 - Discussion - The first paragraph of this section ought to state what the research question was and what was found in this study. However, the authors start this section by discussing socio-demographic characteristics. In my thinking, this study sought to determine the effect of the implementation of the referral guidelines on categories of new admissions, such as walk-ins. The authors should delve into their main findings concerning this question, describe what they found, what other people found concerning the same question and the implications of their findings. If this approach is adopted, all the paragraphs headed "Sociodemographic characteristics of the sample population" should be deleted. The authors could then refine the discussion headed effect of refferral guidelines. The listed recommendations should be deleted and summarized in a statement the conclusions.  

We look forward to receiving your revised manuscript.

Kind regards,

Dickens Otieno Onyango

Academic Editor

PLOS ONE

Journal Requirements:

Reviewers' comments:

Reviewer's Responses to Questions

**Comments to the Author**

1. If the authors have adequately addressed your comments raised in a previous round of review and you feel that this manuscript is now acceptable for publication, you may indicate that here to bypass the “Comments to the Author” section, enter your conflict of interest statement in the “Confidential to Editor” section, and submit your "Accept" recommendation.

Reviewer #1: All comments have been addressed

Reviewer #3: (No Response)

2. Is the manuscript technically sound, and do the data support the conclusions?

Reviewer #1: Yes

Reviewer #3: No

3. Has the statistical analysis been performed appropriately and rigorously? 

Reviewer #1: Yes

Reviewer #3: No

4. Have the authors made all data underlying the findings in their manuscript fully available?

Reviewer #1: Yes

Reviewer #3: Yes

5. Is the manuscript presented in an intelligible fashion and written in standard English?

Reviewer #1: Yes

Reviewer #3: No

6. Review Comments to the Author

Reviewer #1: The author has endeavoured to address all comments and the manuscript is ready for publication.

The language is now more scientific and the methodology section reads well.

Reviewer #3: The study titled "Effect of enforcement of referral guidelines on patterns of orthopedic and trauma admissions to Kenyatta National Hospital Kenya: pre-posttest study design" aimed to evaluate the impact of enforcing referral guidelines on the patterns of orthopedic admissions at Kenyatta National Hospital (KNH) in Kenya. The enforcement of referral guidelines was implemented on July 1, 2021, with the goal of reducing walk-in admissions.

There is very little about trauma admissions in this paper as it seems to focus on the Orthopedic department. For a global audience that understands trauma to be more inclusive than simply orthopedics, I would suggest that the authors remove “trauma” from the title. Or, they could include all admissions involving trauma (craniotomy, laparotomy, thermal injuries, etc.) at KNH.

The study employed a pre- and post-intervention design, comparing data from 459 charts before and 446 charts after the enforcement of national referral guidelines. The authors do not discuss the actual numbers of presentations and admissions during this time period (which would be much more interesting), but instead elected to sample charts during each of the time periods. The findings revealed that the enforcement of referral guidelines to limit walk-ins led to a significant decrease in walk-in admissions (from 54.9% to 45.1%). Additionally, non-trauma orthopedic admissions doubled (from 12.0% to 22.4%), and there was a notable increase in admissions through the Outpatient Clinic and Corporate Outpatient Clinic. The study also observed a decline in emergency admissions and an increase in elective admissions, with the increase in elective cases primarily driven by an increase in female admissions with active insurance cover, tertiary education, non-trauma-related conditions, and older age groups.

It could be interesting for the hospital to look at impact of the enforcement of their policies; however, I do not believe the results reflect the conclusions. The authors have highlighted the enforcement as a positive development. They have limited walk-ins by enforcing a policy to limit walk-ins. There is very little that is novel or interesting to others. However, what they have demonstrated from their results is that access to care is increasingly difficult with the implementation of guidelines. Those who are able to access care now are more likely to be insured, have higher education, and be older. In other words, they have a higher socioeconomic status and can manage the broken system at KNH. So instead of making the system more accessible and appropriate, these guidelines have increased disparities and limited access to Kenyans from all walks of life.

The policy implementation did not increase the number of referral letters. Instead, it increased discrimination to reduce the number of non-insured, younger patients of lower socioeconomic status (with education as a marker). This could be an interesting conclusion to look at the effects of implementation of policy and would be worthy of publication.

The authors blame others for the congestion at KNH without appreciating that delays in care within KNH also lead to delays in discharges and congestion. The high occupancy rates only tell part of the story. What is the length of stay? How quickly are patients operated on (number and percentage within 24 hours)? There appears to be an underlying assumption that no admission to KNH is justified. For example, to turn someone away and require a referral letter from another facility when there is a threat to life or limb, seems inappropriate.

This statement, "This coupled with inadequate staffing levels results in low nurse-patient ratio of 1:10 that compromises not only the quality of nursing care given to patients but also the ability of KNH to effectively perform its statutory obligations” should be supported with evidence/reference.

The authors comment about gender disparities at MTRH. They could also mention the gender disparities that persist outside of the public institutions. At Tenwek Hospital, gender disparities were attributable to delays in definitive care (perhaps because women had more difficulty accessing care at public institutions), but once they arrived to the hospital, the gender disparities in care disappeared suggesting equitable care and inequitable access to care (Otoki K, et al. Gender Disparities in Complications, Costs, and Mortality After Emergency Gastrointestinal Surgery in Kenya. Journal of Surgical Research. 2024 Mar 1;295:846-52.).

The authors mention burn referrals in their references. If the study is examining trauma admissions, patients with burn injuries should be included. If not, trauma should be removed from the title and throughout the manuscript to only focus on orthopedics. For Kenya, some context about the lack of a functional burns referral system may be helpful (Hunter MA, et al. Referral patterns of burn injury in rural Kenya. Journal of Burn Care & Research. 2021 May 1;42(3):454-8.)

The authors mention PCEA Kikuyu. That is an excellent point about how the majority of orthopedics care is delivered by faith-based institutions. The authors should comment about these hospitals provide the majority of care (Parker RK, et al. Surgical training throughout Africa: a review of operative case volumes at multiple training centers. World journal of surgery. 2020 Jul;44:2100-7.) and most cases happen outside of referral centers (Chokotho L, Jacobsen KH, Burgess D, Labib M, Le G, Lavy CB, Pandit H. Trauma and orthopaedic capacity of 267 hospitals in east central and southern Africa. The Lancet. 2015 Apr 27;385:S17.)

The authors should comment on the ethics of turning away patients who have arrived for care. For example, if there is an open fracture and the institution is delaying washout by refusing care, that significantly impacts patient outcomes.

Regarding clarity and consistency, the manuscript would benefit from a thorough revision. For example, the term "pre-posttest study design" is somewhat unconventional; "pre-post intervention study" might be more standard. Additionally, the use of terms like "enforcement of referral guidelines" and "enforcement of the national referral guidelines" should be consistent throughout the paper. Additionally, the manuscript requires careful proofreading for grammatical errors and formatting inconsistencies. For example, there are instances of missing spaces between words, inconsistent use of punctuation, and variations in the formatting of headings and subheadings.

The authors seem to be dismissive of the previous reviewers' comments. Each of these comments should be addressed and the specific place that they are addressed should be included (page/line number) and the specific changes should be included in the comments to reviewers (not just referring to the tracked changes).

7. PLOS authors have the option to publish the peer review history of their article (what does this mean?). If published, this will include your full peer review and any attached files.

Reviewer #1: No

Reviewer #3: No

---

## [Author Response · Author response to Decision Letter 3]

19 Apr 2024

Thanks for reviews. Will be happy to address any additional comments .

Thanks

---

## [Decision Letter · Decision Letter 4]

7 May 2024

Effect of enforcement of the national referral guidelines on patterns of orthopedic admissions to Kenyatta National Hospital, Kenya: Pre-post intervention study

PONE-D-23-24470R4

Dear Dr. Omondi,

We’re pleased to inform you that your manuscript has been judged scientifically suitable for publication and will be formally accepted for publication once it meets all outstanding technical requirements.

Kind regards,

Dickens Otieno Onyango

Academic Editor

PLOS ONE

Additional Editor Comments (optional):

Reviewers' comments:

Reviewer's Responses to Questions

**Comments to the Author**

1. If the authors have adequately addressed your comments raised in a previous round of review and you feel that this manuscript is now acceptable for publication, you may indicate that here to bypass the “Comments to the Author” section, enter your conflict of interest statement in the “Confidential to Editor” section, and submit your "Accept" recommendation.

Reviewer #1: (No Response)

Reviewer #3: All comments have been addressed

2. Is the manuscript technically sound, and do the data support the conclusions?

Reviewer #1: Partly

Reviewer #3: Partly

3. Has the statistical analysis been performed appropriately and rigorously? 

Reviewer #1: Yes

Reviewer #3: Yes

4. Have the authors made all data underlying the findings in their manuscript fully available?

Reviewer #1: Yes

Reviewer #3: Yes

5. Is the manuscript presented in an intelligible fashion and written in standard English?

Reviewer #1: Yes

Reviewer #3: Yes

6. Review Comments to the Author

Reviewer #1: Better grammar and flow of the manuscript

Repetition has been addressed

Manuscript needs another round of proofreading

Study setting – Line 135 to 138 could still be summarized to delve into the orthopaedic details within the shortest time possible

Line 377-384 – the argument may not hold water to say that the increase in female admissions was attributed to a reduction in A&E admissions which were mostly male. If we do admit this argument, we then say that females do not have similar access to admission to KNH and that the enforcement of the guidelines was the game changer. In essence, this may not be true. We could interrogate other possibilities and postulations. But this paragraph requires reconsideration.

Line 457-459 – The argument that the increase in insurance coverage is attributed to increase in female elective admissions may need more qualification and cited evidence on this directly proportional relationship

Reviewer #3: Thank you for addressing the comments.

7. PLOS authors have the option to publish the peer review history of their article (what does this mean?). If published, this will include your full peer review and any attached files.

Reviewer #1: No

Reviewer #3: No
